# Lane-Level Traffic Flow Prediction with Heterogeneous Data and Dynamic Graphs

**Junjie Zhou [1], Siyue Shuai [2], Lingyun Wang [2], Kaifeng Yu [2], Xiangjie Kong [2,*], Zuhua Xu [1] and Zhijiang Shao [1]**

[1] College of Control Science and Engineering, Zhejiang University, Hangzhou 310027, China; 12132091@zju.edu.cn (J.Z.); zhxu@zju.edu.cn (Z.X.); szj@zju.edu.cn (Z.S.)

[2] College of Computer Science and Technology, Zhejiang University of Technology, Hangzhou 310023, China; siyueshuai@outlook.com (S.S.); kuluo_wly@outlook.com (L.W.); kfyu1996@outlook.com (K.Y.)

\* Correspondence: xjkong@ieee.org; Tel.: +86-158-4090-1926

**Abstract:** With the continuous development of smart cities, intelligent transportation systems (ITSs) have ushered in many breakthroughs and upgrades. As a solid foundation for an ITS, traffic flow prediction effectively helps the city to better manage intricate traffic flow. However, existing traffic flow prediction methods such as temporal graph convolutional networks(T-GCNs) ignore the dissimilarities between lanes. Thus, they cannot provide more specific information regarding predictions such as dynamic changes in traffic flow direction and deeper lane relationships. With the upgrading of intersection sensors, more and more intersection lanes are equipped with intersection sensors to detect vehicle information all day long. These spatio-temporal data help researchers refine the focus of traffic prediction research down to the lane level. More accurate and detailed data mean that it is more difficult to mine the spatio-temporal correlations between data, and modeling heterogeneous data becomes more challenging. In order to deal with these problems, we propose a heterogeneous graph convolution model based on dynamic graph generation. The model consists of three components. The internal graph convolution network captures the real-time spatial dependency between lanes in terms of generated dynamic graphs. The external heterogeneous data fusion network comprehensively considers other parameters such as lane speed, lane occupancy, and weather conditions. The codec neural network utilizes a temporal attention mechanism to capture the deep temporal dependency. We test the performance of this model based on two real-world datasets, and extensive comparative experiments indicate that the proposed heterogeneous graph convolution model can improve the prediction accuracy.

**Keywords:** lane-level traffic flow prediction; temporal–spatial correlation; graph convolution neural network; attention mechanism

## 1. Introduction

As a significant part of smart city construction, an intelligent transportation system (ITS) acts as a powerful tool to enhance the quality of city transportation. Benefiting from the rapid development of the Internet of Things [1], the ITS can provide key information to aid in managing traffic through the connection between traffic sensors and the Internet. For instance, traffic police can inspect illegal parking and allocate traffic resources more efficiently with the deployment of automatic license plate recognition (LPR) infrastructure [2] or identify potentially illegal or dangerous driving behaviors by assessing driving style [3]. Moreover, with the vigorous development of mobile applications and intelligent vehicle terminals, every driver can easily obtain information about traffic congestion location predictions [4] and recommended driving routes. In addition, the traditional service industry relying on the road network has also been benefited in a variety of ways, such as the birth of crowdsourced logistics in the express industry, which could coordinate taxi passenger

and express delivery services [5], and the location planning of business districts in the urban layout [6]. Therefore, ITS benefits not only traffic managers but also all participants including drivers, pedestrians, and so on. For the purpose of strengthening the foundation of ITSs, more researchers have funneled study into the field of traffic flow prediction.

Based on different methods, traffic flow prediction analyzes and generalizes the traffic characteristics of both common and special areas (e.g., schools and hospitals) in both general and special periods (e.g., during congestion or after accidents) and generates accurate predictions of future traffic states [7]. Current traffic prediction methods fall into two types: model-driven methods and data-driven methods [8–10]. However, the former methods have the disadvantage of being unable to handle cases affected by special factors or events such as weather conditions and traffic accidents.

Thus, in order to take special situations into account, current studies mainly adopt data-driven deep learning methods, which are capable of analyzing multi-dimensional data. Lint et al. [11] modeled time series with a recurrent neural network (RNN). However, the RNN has the defects of gradient disappearance and long-term dependence, and therefore Ma et al. [12] used a long short-term memory (LSTM) method to conduct a predictive analysis of the data collected from microwave sensors. By making use of a gate recurrent unit (GRU), an effective variant of LSTM, Fu et al. [13] improved the speed forecast effectiveness. The above three studies only focused on either the temporal dependence or spatial dependence; they did not consider both comprehensively. Although Yao et al. [14] focused on temporal–spatial correlation, and the proposed spatial–temporal dynamic network (STDN) model combined the dynamic change in spatial correlation with the periodicity of time correlation, Yao's study had the same disadvantage as the previous research, i.e., lanes at an intersection were considered as a whole and their differences were not distinguished [15].

At different intersections and in different flow directions, owing to the high mobility of vehicles, lanes often show diverse traffic patterns [15,16]. As shown in Figure 1, in the morning, represented by red arrows, the traffic flow directed from right to left has a larger scale, and the opposite is true in the evening peak. In view of the ever-increasing development of smart cities, traffic flow prediction technology should detect dynamic changes in lanes more quickly and efficiently, with multiple trajectory data. Accordingly, from a data-driven perspective, a technology of traffic prediction at the lane level emerges and becomes one of the most vital parts of an ITS.

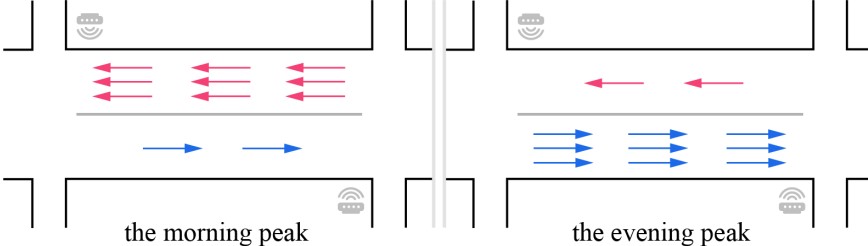

**Figure 1.** Different traffic conditions of lanes in morning and evening peaks.

The lane-level traffic prediction method has attracted the attention of researchers mainly because it can be applied to a variety of complex dynamic prediction scenarios. Based on the improvement of the existing prediction method, it could play a huge role in the future in applications such as unmanned driving, lane-level positioning [17], and cloud video surveillance traffic detection [18].

In order to focus on the temporal and spatial characteristics of traffic flow, Gu et al. [19] used an entropy-based gray correlation method to analyze the correlation between lanes and an LSTM-GRU model to derive the spatial–temporal information matrix which is iterated to output the speed prediction. Ke et al. [20] proposed an innovative two-stream multichannel convolutional neural network (TMCNN) model, based on the view that different lanes at an intersection can be equated with diverse channels in an image. By introducing a theory of lane-changing behavior, Xie et al. [21] established a model for lane-changing

and its implementation based on a deep belief network (DBN) and LSTM. In addition, Lu et al. [22] and Ma et al. [23] utilized a convolutional LSTM network (ConvlTM) model to process temporal–spatial information.

All the above authors carried out in-depth studies on the temporal and spatial characteristics of traffic flow, but excessive emphasis on these characteristics will inevitably lead to the neglect of other heterogeneous data; for example, some time-varying characteristics are not taken into account. Consequently, in our work, considering static and dynamic road network structures, we propose a heterogeneous graph convolution model that takes advantage of the road occupancy rate, minimum visibility, and other data to minimize the deviations in prediction results caused by a single data source.

Inspired by some of the latest research on dynamic graphs [24,25], we propose generating dynamic graphs in an internal graph convolution network to determine the correlation between lanes. Since the impact of weather conditions and road occupancy on flow prediction cannot be ignored, we fuse other heterogeneous data in an external data fusion network. In addition, in order to mine the time-dependence of traffic flow, we deploy a codec network with a temporal mechanism.

In summary, regarding lane-level prediction, this study mainly demonstrates innovation and progress in the following three respects:

- We propose a heterogeneous graph convolution model based on dynamic graph generation. In order to discover the spatial dependence of dynamic graphs, a graph convolutional network (GCN) is introduced into this model, and a temporal attention mechanism is applied to further study the time-dependence.
- We analyze other heterogeneous data to obtain more potential information that has not been previously considered. That is, by incorporating multivariate data such as occupancy and minimum visibility into heterogeneous data fusion networks, we can reduce the loss of accuracy.
- We conduct extensive experiments on two real-world datasets, and the results show that our proposed model can increase the precision of lane-level traffic state prediction.

## 2. Methodology

Figure 2 exhibits the model architecture of the heterogeneous graph convolutional network. The network consists of three main components: (1) an internal graph convolution network based on dynamic graphs; (2) an external heterogeneous data fusion network; and (3) a codec network based on an attention mechanism. The internal convolution network is integrated with the external heterogeneous data fusion network, and they are represented as an association convolution block (ACB). Here, $\tilde{x}_T^{sp}$ represents the output of the internal graph convolution network and $\tilde{x}_T^{out}$ represents the final result of the external heterogeneous data fusion network.

The process shown in the whole frame starts from the internal convolution network of the ACB, i.e., the left part of the ACB shown in Figure 2. The participating lanes are divided into two categories according to their relationship with the target lane: (1) for lanes at different intersections, the internal convolution network combines a dynamic graph matrix and a static adjacency matrix and (2) for lanes at the same intersection, only a dynamic graph matrix is generated, with no static adjacency matrix. Graph convolutional operations are performed according to the above conditions, and the outputs are fused into the final internal graph convolution result.

Regarding the external fusion network of the ACB, by adopting an attention mechanism, reasonable weights are allocated to the various parameters and then the results are fused with the convolution result of the internal graph to produce the outputs of the ACB.

In the last part of the whole framework, the encoder assigns the temporal weights to the hidden layers in the GRU units by utilizing the attention mechanism. After obtaining the context vector constructed in the encoder, the decoder outputs the final prediction result through iterative processing.

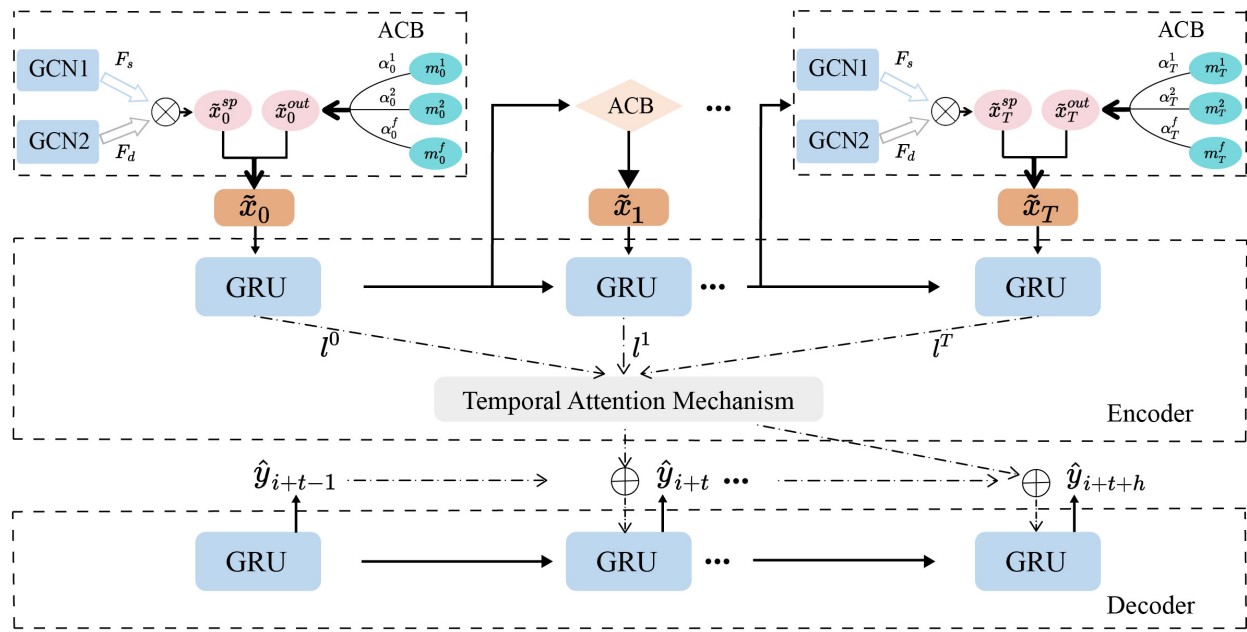

**Figure 2.** Architecture of the heterogeneous graph convolution model.

### 2.1. Internal Graph Convolution Network

Since traffic information can be regarded as graph signals, many researchers apply graph neural networks in traffic flow prediction to capture spatial characteristics in road networks. In this paper, we utilize the graph convolution network based on a spectral domain to mine the deep spatial dependency in static adjacency graphs and dynamic graphs.

First, according to the road topology, we define an undirected and weighted graph $G = (V, E)$, where $V = \{v_1, v_2, \cdots, v_N\}$ is the set of nodes, $N$ is the number of nodes, and $E$ is the set of links, representing the connectivity between nodes. The weights and static adjacency matrix of $G$ are calculated as follows:

$$\mathbf{A^I_{i,j}} = \mathbf{A^I_{j,i}} = \begin{cases} \frac{1}{d_{i,j}}, & d_{i,j} \neq 0 \\ 0, & d_{i,j} = 0 \end{cases} \tag{1}$$

where $d_{ij}$ denotes the distance between $i$ and $j$. $\mathbf{A^I} \in R^{N \times N}$ is the static adjacency matrix, which has only two types of values: zero represents the fact that two nodes do not correlate and a non-zero value represents the correlation degree (i.e., the weight) between two nodes. Thus, $\mathbf{A^I}$ means that the smaller the distance between two nodes, the greater the correlation degree.

We apply Chebyshev polynomials to greatly reduce the operation convolution time:

$$g_\theta(\Lambda) = \sum_{k=0}^{K-1} \partial_k T_k(\tilde{\Lambda}) \tag{2}$$

where $\partial_k$ is the coefficient of the Chebyshev polynomials and $\tilde{\Lambda}$ is the maximum eigenvalue of the Laplace matrix. The recursive form of Chebyshev polynomials can be expressed as $T_k(x) = 2T_{k-1}(x) - T_{k-2}(x)$, where $T_0 = 1$ and $T_1 = x$.

Graph convolution networks based on Chebyshev polynomials help to improve the efficiency but cannot sense dynamic changes in traffic patterns through the static adjacency matrix in Equation (1). Taking the change in traffic flow in the morning and evening peaks as an example, during the early peak, the traffic flow on main roads between the suburbs and cities tends to increase sharply, and the traffic direction is from the suburbs to the cities. In contrast, during the evening peak, the number of vehicles driving from cities to the suburbs shows a growth trend. The static adjacency matrix only focuses on the influence of

the distance between two nodes and ignores the dynamics caused by the change in vehicle flow at different times. Figure 3 shows the correlation degree between nodes changing over time, where different colors denote different correlation degrees.

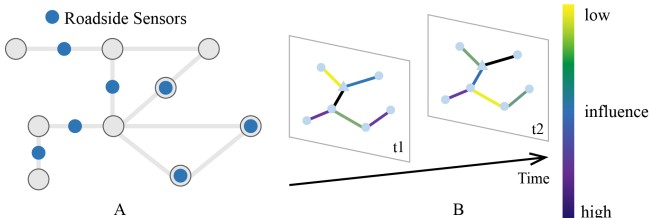

**Figure 3.** Association between nodes. Subfigure (**A**) shows the distribution of each sensor in a road network. Subfigure (**B**) shows the correlation degree changing over time.

Apart from ignoring temporal variability, Equation (1) also overlooks the impact of similar traffic patterns between nodes. As shown in Figure 4, S3 is closest to S1, S2, and S4, and in consequence these links have higher weights in subfigure A. We assume that after the analysis of traffic patterns, the pattern of S3 is the most similar to that S6, i.e., their traffic flows are related during the morning and evening peaks, and congestion at S6 will spread to S3 in a short time. Then, we are able to draw the dynamic graph in subfigure B. Although S3 is far away from S6, they have the highest link weight.

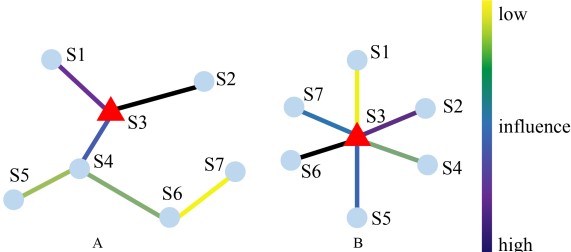

**Figure 4.** Static and dynamic links between nodes. Subfigure (**A**) shows the static graph calculated by node distances. Subfigure (**B**) shows the dynamic graph calculated by similarity of traffic patterns.

Considering the dynamic changes with time and traffic patterns, we innovatively propose the internal graph convolutional network based on dynamic graph generation. Figure 5 shows the architecture of the network.

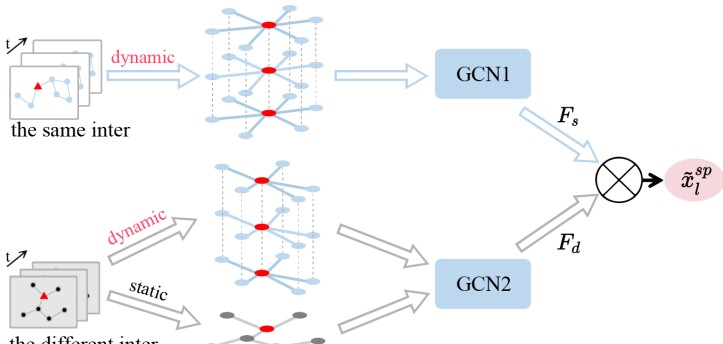

**Figure 5.** Internal graph convolutional network based on dynamic graph generation.

Before model training, in terms of the adjacency relationship with the target lane, we divide all participating lanes into same-intersection adjacency lanes $X_s$ and different-intersection adjacency lanes $X_d$. Traffic flow parameters have the ability to reflect the congestion degree of the road network and the probability of accidents occurring. Therefore,

we select the traffic flow as the metric and calculate the correlation degree of each node, using the Pearson correlation coefficient to generate the dynamic graph:

$$p_{i,j}^s = \frac{Z \sum x_l^{s,i} x_l^{s,j} - \sum x_l^{s,i} \sum x_l^{s,j}}{\sqrt{Z \sum (x_l^{s,i})^2 - (\sum x_l^{s,i})^2} \sqrt{Z \sum (x_l^{s,j})^2 - (\sum x_l^{s,j})^2}} \tag{3}$$

$$\mathbf{A}_{i,j}^{s,c} = \mathbf{A}_{j,i}^{s,c} = \begin{cases} p_{i,j}^s & , p_{i,j}^s \neq 0 \\ 0 & , p_{i,j}^s = 0 \end{cases} \tag{4}$$

Equation (3) shows the formula for the Pearson correlation coefficient, where $x_l^{s,i}$ and $x_l^{s,j}$ are the parameters of lane $i$ and lane $j$ at moment $l$, respectively, and $Z$ is the amount of data for each lane. $A^{s,c}$ in Equation (4) is the dynamic graph matrix of the same-intersection lanes and $p_{i,j}^s$ is the correlation degree between lane $i$ and lane $j$ at the same intersection. Similarly, we can derive the dynamic graph matrix $A^{d,c}$ for different-intersection lanes.

When performing a graph convolution operation on same-intersection lanes, we utilize the dynamic matrix $A^{s,c}$ to generate a convolution kernel $g_\theta^{s,dy}(\Lambda)$ and convolute $X_s^l$ at moment $l$:

$$F_s = g_\theta^{s,dy}(\Lambda) X_l^s = \sum_{k=0}^{K-1} \partial_k T_k^{s,dy}(\tilde{\Lambda}) X_l^s \tag{5}$$

When performing a graph convolution operation on different-intersection lanes, we utilize the convolutional kernels $g_\theta^{d,dy}(\Lambda)$ and $g_\theta^{d,st}(\Lambda)$ generated by the dynamic matrix $A^{d,c}$ and the static adjacency matrix, respectively, to convolute $X_d^l$:

$$F_d = \left( (g_\theta^{d,dy}(\Lambda) + g_\theta^{d,st}(\Lambda)) X_l^d \right) \odot W \tag{6}$$

$$g_\theta^{d,dy}(\Lambda) + g_\theta^{d,st}(\Lambda) = \sum_{k=0}^{K-1} \partial_k \left( T_k^{d,st}(\tilde{\Lambda}) + T_k^{d,st}(\tilde{\Lambda}) \right) \tag{7}$$

where $W$ is the trainable weight matrix and $\odot$ denotes the element-wise product.

The final result of the internal graph convolution network is calculated by the fusion of $F_s$ and $F_d$ at moment l:

$$\tilde{x}_l^{sp} = relu \left( F_s * W_l^{sp} + b_l^{sp} + (F_d * V_l^{sp} + c) \right) \tag{8}$$

where $W_l^{sp}$ and $V_l^{sp}$ are convolution kernels and $b_l^{sp}$ and $c$ are biases corresponding to the kernels.

### 2.2. External Heterogeneous Data Fusion Network

Through previous experiments, it was found that one of the main reasons for the low accuracy of traffic flow prediction results is the fact that multi-source heterogeneous data are left out of consideration and data of only one type are acquired. According to the speed–flow equation for traffic, the nonlinear relationship between speed, flow, and road occupancy rate is so complex that we cannot confirm one variable by any other single variable. For instance, when at a flat peak, the speed of the vehicle is high but the road occupancy is low, so the flow is quite low. However, in traffic congestion, though the occupancy rate is high, the speed stays at a low level and the flow remains at a small value. Therefore, the current traffic flow condition cannot be analyzed according to only one aspect of the traffic status, as the forecast result will have a large initial difference from the actual data.

Furthermore, weather is a significant factor that affects the traffic conditions to a great extent. Whether the weather is sunny or not affects the driving safety factor, which leads to higher vehicle speeds when the weather is sunny, and the saturation flow of the lane can

be higher at this time. In the case of bad weather such as snowstorms and fog, the driver takes the initiative to slow down due to the low visibility, and reaching the saturation level is unlikely.

The above shows that there are many determinants of traffic conditions. Considering only single-source traffic flow parameters leads to the traffic conditions being oversimplified and the complexity being ignored. To be rigorous, the model requires vehicle speed, road occupancy, minimum visibility, and other data as a supplement.

We apply an external heterogeneous data fusion network to solve the problem of analysis error caused by a lack of information. Figure 6 shows the architecture of the network. The neural network adopts an attention mechanism and reasonably allocates the weights of various heterogeneous characteristic parameters.

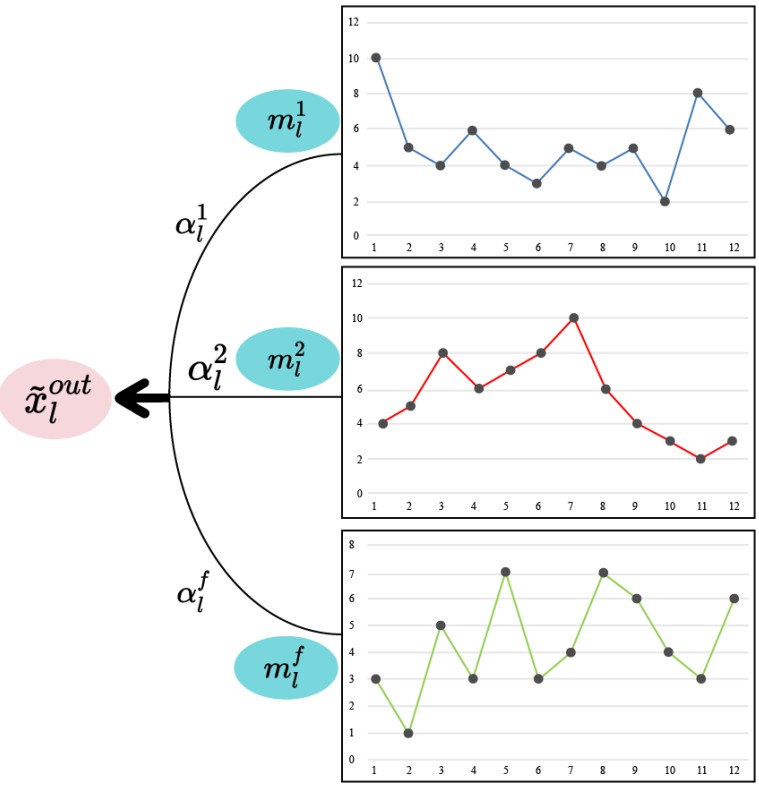

**Figure 6.** External heterogeneous data fusion network.

Attention scores are calculated as:

$$e_l^p = V_l^{in} \cdot \sigma(W_{l1}^{in} \cdot [d_{l-1}^{en}; s_{l-1}^{en}] + W_{l2}^{in} \cdot m_l^p + b_l^{in}) \tag{9}$$

$$\alpha_l^p = \frac{\exp(e_l^p)}{\sum_{j=1}^{f} \exp(e_l^j)} \tag{10}$$

where $W_{l1}^{in}$, $W_{l2}^{in}$, and $V_l^{in}$ are trainable weight matrices, $b_l^{in}$ is the bias, $[d_{l-1}^{en}; s_{l-1}^{en}]$ is the result of connecting hidden layers with cell state layers in the GRU units of the decoder, $m_l^p$ is the $p$-th heterogeneous characteristic parameter at moment $l$, and $a_l^p$ is the allocated weight. The final result $\tilde{x}_l^{out}$ is calculated as:

$$\tilde{x}_l^{out} = \left(\alpha_l^1 m_l^1, \alpha_l^2 m_l^2, \ldots, \alpha_l^f m_l^f\right)^{\mathrm{T}} \tag{11}$$

### 2.3. Codec Network Based on Attention Mechanism

As shown in Figure 7, the final result of the ACB is fused with the internal and external networks:

$$\tilde{x}_l = \left[ W_{l1} \cdot \tilde{x}_l^{sp} + b_{l1}; W_{l2} \cdot \tilde{x}_l^{out} + b_{l2} \right] \tag{12}$$

where $W_{l1}$ and $W_{l2}$ are trainable weight matrices and $b_{l1}$ and $b_{l2}$ are biases corresponding to the matrices.

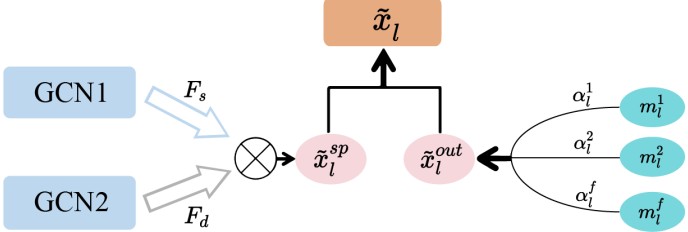

**Figure 7.** Fusion of internal network and external network.

In spite of the deep spatial dependency between nodes in the result of the ACB, temporal dependency must be further mined to improve prediction accuracy. In traffic flow prediction, extensive experimental studies have proved that LSTM and GRU models have the advantage of effectively processing time series by collecting historical information and referring to previous scenarios. Based on this conclusion, in order to ensure our model obtains this advantage, we construct the codec network with the GRU. Due to the limited storage space, the length of the context vectors is fixed, which restricts the expression ability of the codec network.

For the benefit of finding long-term dependencies in sequences, we combine the codec network with a temporal attention mechanism. The temporal attention mechanism allocates reasonable weights to each period efficiently via self-learning and parallel computing. Temporal attention weights are calculated as follows:

$$k^p = V_l^{en} \cdot \sigma \left( W_{l1}^{en} \cdot [d_{l-1}^{en}; s_{l-1}^{en}] + W_{l2}^{en} \cdot \tilde{x}_l + b_l^{en} \right) \tag{13}$$

$$\ell^p = \frac{\exp(k^p)}{\sum_{i=1}^{T} \exp(k^i)} \tag{14}$$

where $W_{l1}^{en}$, $W_{l2}^{en}$, and $V_l^{en}$ are trainable weight matrices, $[d_{l-1}^{en}; s_{l-1}^{en}]$ is the result of connecting hidden layers with cell state layers in the previous GRU units of the decoder, $\tilde{x}_l$ is the output of the ACB at moment $l$, and $b_l^{en}$ is the bias. Then, the context vectors are calculated as:

$$h_l^{en} = G^{en} \left( h_{l-1}^{en}, \tilde{x}_l \right) \tag{15}$$

$$C = \sum_{l=1}^{T} \ell^l \cdot h_l^{en} \tag{16}$$

where Equation (15) derives the output $h_l^{en}$ of the GRU units at moment $l$, $G^{en}$ denotes the GRU units of the encoder, $h_{l-1}^{en}$ is the output of the previous GRU unit, and $\tilde{x}_l$ is the output of the ACB at moment $l$. Equation (16) derives the context vector $C$, and $\ell^l$ is the temporal weight.

Then, the context vectors are decoded as follows:

$$d_l^{de} = G^{de} \left( d_{l-1}^{de}, [\hat{y}_{l-1}; C] \right) \tag{17}$$

$$\hat{y}_l = V_l^{de} \cdot \sigma \left( W_l^{de} \cdot [d_l^{de}; C] + b_l^{de} \right) \tag{18}$$

where $G^{de}$ denotes the GRU unit of the encoder, $d^{de}_{l-1}$ is the previous hidden layer of the decoder, $d^{de}_l$ is the current hidden layer of the decoder, $W^{de}_l$ and $V^{de}_l$ are trainable matrices, $b^{en}_l$ is the bias, and $\hat{y}_l$ is the final prediction result.

## 3. Experiments and Results

### 3.1. Experimental Data

Aiming at conducting a dynamic analysis, we performed prediction tests on heterogeneous data, including traffic flow data and weather information as follows:

- Traffic data: PEMSD4 and PEMSD8 are two real-world datasets from the Caltrans Performance Measurement System (PeMS). The detailed information is shown in Table 1.
- Weather information: We collected weather data for the San Francisco Bay area and for San Bernardino during the periods shown, selected the visibility metric, and then integrated the data into the test as a special factor.

In order to facilitate the test in the internal convolution network of the ACB, we preprocessed the data in advance. First, we classified the data according to whether they referred to the same intersection or the same lane. Then, we obtained the total traffic flow, average occupancy rate of lanes, and average speed every five minutes. According to the road speed limit and actual situations, we removed unreasonable data, smoothed burr points, and utilized the sliding window method with fixed length to estimate missing data. To verify the comparison of the accuracy of traffic flow prediction, we split each dataset and selected 47 days as the training set and 15 days as the test set.

**Table 1.** Traffic data details.

|  | Area | Sensors | Time Range |
|---|---|---|---|
| PeMSD4 | 29 roads of the San Francisco Bay | 307 sensors | 56 days: 1 January 2018—28 February 2018 |
| PeMSD8 | 8 roads of the San Bernardino | 170 sensors | 62 days: 1 July 2016—31 August 2018 |

### 3.2. Prediction Task

In the preprocessing of the dataset, the traffic flow and weather data were recorded every five minutes. Our prediction task was to use the first five records to predict the sixth datum. That is, the traffic flow datum for the next five minutes was predicted based on the information from the first sixty minutes of data. Finally, our proposed lane-level traffic flow prediction algorithm was evaluated by comparing the real data with the predicted data.

### 3.3. Experimental Setting

To demonstrate the superiority of the proposed model, we compared it with seven other algorithms, including parametric methods, non-parametric methods, and deep learning methods:

- HA (historical average): this method is based on statistics and calculates the average value of traffic flow parameters to derive the prediction results.
- SVM (support vector machines): this is a supervised learning method and classifies data by binary methods. The optimal hyperplane can maximize the distance between multiple categories.
- ARIMA (autoregressive integrated moving average): this is a time series prediction method, composed of an autoregressive block and a moving average block.
- GRU (gate recurrent unit): this model is a variant of LSTM that removes forgetting gates and only consists of update gates and reset gates. It requires fewer parameters and converges more easily.
- T-GCN [26]: this model is a time series neural network with a GRU-GCN structure, where the GCN deals with spatial dependency and the GRU deals with temporal dependency.

- ST-AFN [27]: this model represents our previous work and mainly has a temporal–spatial attention mechanism construction. The network consists of a speed processing network, a spatial encoder, and a temporal decoder.
- ASTGCN [28]: this model is graph convolution network that divides a temporal sequence into three parts, i.e., recent, daily-periodic and weekly-periodic dependencies. Prediction results are derived by fusing the outputs of the three parts.

All experiments were run on the same computer, where the OS was Ubuntu 18.04, the memory capacity was 64 GB, the CPU was an Intel Xeon Silver, and the GPU was an NVIDIA Quadro M4000. We used Pytorch 1.6.0 and set a series of hyperparameters as follows: 0.0001 for the learning rate; 128 for the batch size; 64 for the size of the hidden layers and cell states in the GRU; and 12 for the step length for historical data. Moreover, we selected the mean absolute error (MAE), the root mean squared error (RMSE), and the mean absolute percentage error (MAPE) to evaluate the prediction performance.

### 3.4. Performance Comparison

First, we compared the accuracy of our proposed algorithm with seven other prediction algorithms. Then, a comparison experiment was conducted considering the difference in traffic flow characteristics between weekdays and weekends. After verifying the effectiveness of our proposed algorithm, ablation experiments were conducted in order to investigate whether the internal graph convolutional network or the external heterogeneous data fusion network in the algorithm played a more important role. Finally, the time consumption of our algorithm was compared with three other algorithms.

### 3.4.1. Overall Comparison

Table 2 shows the results of the overall performance comparison. The smaller the values of MAE, RMSE, and MAPE, the better the prediction effect. Although the HA method based on statistics showed higher efficiency than the other methods and the evaluation indicators were not all larger than the others, this simple algorithm has an inability to process complex, changeable, real-time traffic flow data. The ARIMA method focuses on the regularity and stability of traffic flow data in the time dimension, and therefore it performed better than the traditional machine learning method SVM, but the accuracy was still low. The performance of GRU was superior to ARIMA, but both of them lack spatial dependency, which is exactly what T-GCN has to help it perform better. Although ST-AFN and ASTGCN both take the spatio-temporal characteristics of the data into account, the latter method is more accurate for constructing a multi-branched structure. The results show that the proposed heterogeneous graph convolution network performs best overall and that that MAE, RMSE, and MAPE reached 16.1654, 24.656, and 12.413% in PeMSD4 and 15.486, 22.513, and 13.641% in PeMSD8, respectively.

**Table 2.** Overall performance comparison.

| | PeMSD4 | | | PeMSD8 | | |
|---|---|---|---|---|---|---|
| | **MAE** | **RMSE** | **MAPE** | **MAE** | **RMSE** | **MAPE** |
| HA | 35.428 | 51.144 | 25.156% | 28.438 | 48.367 | 30.579% |
| SVM | 42.924 | 48.843 | 22.511% | 30.546 | 48.645 | 28.647% |
| ARIMA | 32.540 | 42.596 | 20.168% | 33.726 | 45.286 | 25.213% |
| GRU | 29.120 | 40.603 | 18.551% | 28.863 | 41.565 | 27.226% |
| T-GCN | 25.842 | 31.457 | 14.657% | 24.643 | 30.665 | 15.691% |
| ST-AFN | 22.293 | 28.032 | 16.195% | 21.156 | 25.562 | 18.532% |
| ASTGCN | 19.531 | 26.924 | 14.522% | 18.685 | 27.651 | 16.650% |
| **Ours** | **16.165** | **24.656** | **12.413%** | **15.486** | **22.513** | **13.641%** |

In order to visualize the difference between the test performances of different methods, we constructed a bar chart and box chart according to Table 2, as shown in Figure 8. Through detailed analysis and graphical presentation of the experimental results, it can be

determined that the the new proposed algorithm outperformed the other seven methods in the overall performance of the prediction task.

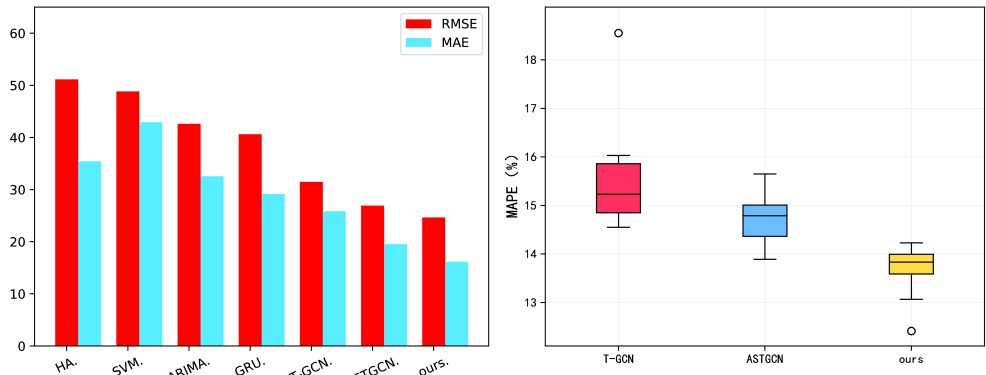

**Figure 8.** Performance comparison charts. The left subfigure is the bar chart, and the right subfigure is the box chart.

### 3.4.2. Comparison of Weekends and Workdays

As a result of people's different travel intentions, the fluctuation range of traffic flow is more obvious on weekdays than on weekends. The main difference is that traffic flow increases more rapidly during the morning and evening peaks on weekdays. In order to compare the differences between algorithms regarding the prediction for these two periods, we divided the prediction results into working days and non-working days for the statistics. Table 3 shows the metrics of T-GCN, ASTGCN, and the heterogeneous graph convolution network for workdays and weekends. The prediction results of all these three algorithms for non-working days were better than those for working days. Among them, our proposed model reached the optimal prediction results, and the accuracy for non-working days was higher than that for working days.

**Table 3.** Performance comparison of working days and nonworking days.

|  | **Working Days** | | | **Non-Working Days** | | |
|---|---|---|---|---|---|---|
|  | **MAE** | **RMSE** | **MAPE** | **MAE** | **RMSE** | **MAPE** |
| T-GCN | 27.546 | 32.135 | 16.165% | 23.348 | 30.398 | 14.418% |
| ASTGCN | 20.125 | 25.972 | 17.214% | 17.835 | 28.142 | 14.186% |
| **Ours** | **17.981** | **25.031** | **14.155%** | **14.013** | **22.489** | **12.410%** |

In order to better reveal the prediction accuracy and distinction of the heterogeneous graph convolution model on different types of days, we selected a working day from PeMSD4 and a non-working day from PeMSD8 for in-depth analysis. As shown in Figures 9 and 10, the heterogeneous graph convolution model obtained good traffic flow prediction results. It can be seen from the curve overlap degree of Figures 9 and 10 that the prediction effect for non-working days was slightly better than that for working days. The prediction effect in the morning and evening peaks was not good enough, which means that stability of traffic conditions also affects the final prediction results, i.e., the prediction accuracy is high when the traffic conditions are relatively stable but the gap between the prediction result and the real value becomes large when fluctuation occurs in the traffic flow. In addition, it can be observed that the prediction error at noon was large, as the fluctuations and interleaving of the curves are quite different. Furthermore, the model has the defects of failing to predict the maximum flow value in the morning and evening peaks.

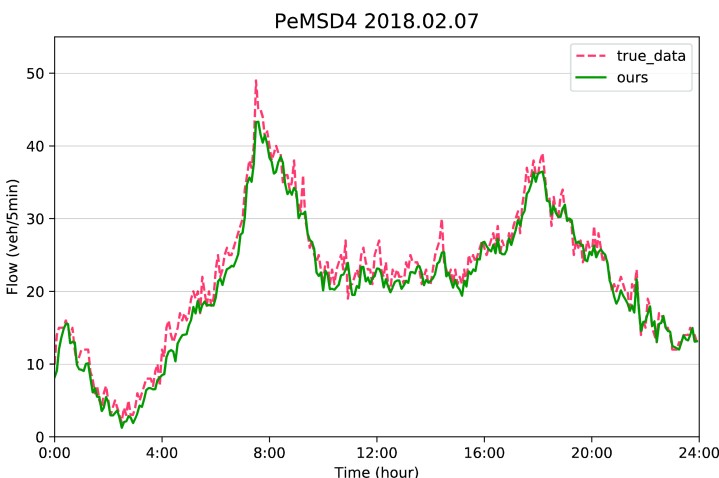

**Figure 9.** Prediction results for the working day in PeMSD4.

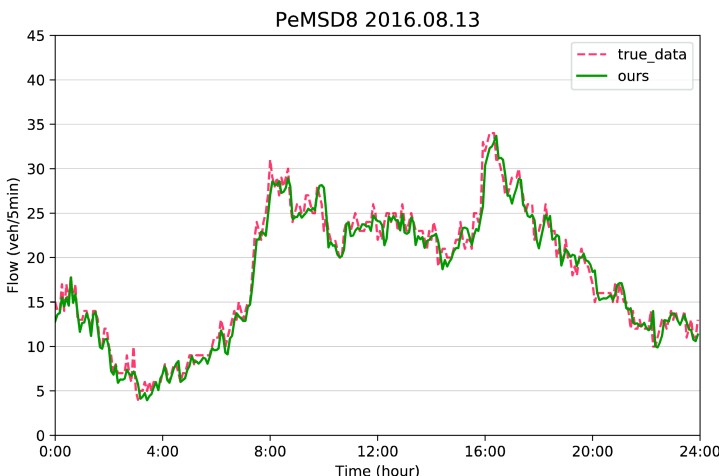

**Figure 10.** Prediction results for the non-working day in PeMSD8.

### 3.4.3. Ablation Comparison

To determine whether the internal graph convolutional network or the external heterogeneous data fusion network played a key role in the heterogeneous graph convolution network in improving the accuracy, we conducted ablation experiments by removing a certain portion of the proposed network. The experimental results are shown in Table 4, where Ours* is the prediction result of the static graph convolution network without the internal graph convolution network and Ours** is the prediction result without the external heterogeneous data fusion network. Since the accuracies of Ours** and Ours* are similar and much lower than for the full structure, this indicates that both structures play an important role in the whole framework, and they are both indispensable in improving the accuracy.

**Table 4.** Performance comparison for ablation experiment.

| | PeMSD4 | | | PeMSD8 | | |
|---|---|---|---|---|---|---|
| | **MAE** | **RMSE** | **MAPE** | **MAE** | **RMSE** | **MAPE** |
| Ours** | 26.221 | 37.542 | 14.468% | 26.462 | 38.403 | 19.135% |
| Ours* | 25.437 | 35.416 | 16.105% | 25.154 | 35.168 | 20.896% |
| T-GCN | 25.842 | 31.457 | 14.657% | 24.643 | 30.665 | 15.691% |
| ASTGCN | 19.531 | 26.924 | 14.522% | 18.685 | 27.651 | 16.650% |
| **Ours** | **16.165** | **24.656** | **12.413%** | **15.486** | **22.513** | **13.641%** |

### 3.4.4. Training Cost Comparison

As well as focusing on the accuracy of the algorithms, we compared the training costs as an indicator of the computational efficiency of T-GCN, ASTGCN, ST-AFN, and the heterogeneous graph convolution network. As shown in Table 5, the ST-AFN method showed the fastest training speed due to its efficient attention mechanism, and ASTGCN, with a multi-branched network, took the most time. Since the network structure of T-GCN is relatively simple, it also performed well with respect to time cost, though it was slightly inferior to ST-AFN. Regarding the heterogeneous graph convolution network, the performance was also good in terms of time consumption, and not too much time was expended.

**Table 5.** Training cost comparison.

|         | PeMSD4    | PeMSD8    |
| ------- | --------- | --------- |
| ST-AFN  | 182.17 s  | 158.29 s  |
| ASTGCN  | 310.65 s  | 275.12 s  |
| Ours    | 276.48 s  | 220.14 s  |
| T-GCN   | 238.26 s  | 194.31 s  |

## 4. Discussion

As smart cities are developing by leaps and bounds, traffic flow management is required to improve prediction accuracy. Therefore, researchers have intensified research into lane-level traffic flow prediction, as it can capture urban road conditions and dynamic changes in the objective environment more accurately than existing prediction methods. We proposed a heterogeneous graph convolution network based on dynamic graph generation, which generally adopts the codec structure. The ACB consists of the internal graph convolution network and external heterogeneous data fusion network, fusing the outputs of both networks. The encoder allocates reasonable weights to each period through a temporal attention mechanism to construct context vectors. Finally, the decoder decodes context vectors and implements the traffic flow prediction. We experimented with two real-world datasets from PeMS and demonstrated the effectiveness of the proposed network. Although the codec structure and attention mechanism improved the efficiency, the network still had defects with respect to real-time computing in changeable traffic conditions. In order to better coordinate the overall framework, we will probe more deeply into edge computing methods, to unload heavy computing tasks to the road-side unit (RSU) and enhance the efficiency of data transmission to improve the quality of service [29,30]. In addition, the idea of model pre-training in transfer learning also provides a new direction for improving real-time learning effectively in the future.

**Author Contributions:** Conceptualization, methodology, and project administration, J.Z.; experiments and analysis, J.Z., L.W., K.Y. and S.S.; investigation, S.S. and L.W.; resources and data curation, J.Z. and K.Y.; writing—original draft preparation, J.Z., K.Y., L.W. and S.S.; writing—review and editing, X.K., Z.X. and Z.S.; supervision, X.K.; funding acquisition, X.K. and J.Z. All authors have read and agreed to the published version of the manuscript.

**Funding:** This work was supported in part by the "Pioneer" and "Leading Goose" R&D Programs of Zhejiang under Grant 2022C01050, in part by the National Natural Science Foundation of China under Grant 62072409, in part by the Zhejiang Provincial Natural Science Foundation under Grant LR21F020003, and in part by the Fundamental Research Funds for the Provincial Universities of Zhejiang under Grant RF-B2020001.

**Institutional Review Board Statement:** Not applicable.

**Informed Consent Statement:** Not applicable.

**Data Availability Statement:** Not applicable.

**Conflicts of Interest:** The authors declare no conflict of interest.

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
