# Peer review of "Lane-Level Traffic Flow Prediction with Heterogeneous Data and Dynamic Graphs"

_applsci, doi:10.3390/app12115340_

Round 1

Reviewer 1 Report

I liked the article because I conducted similar studies with my co-authors in 2018. But what confuses me is the serious discrepancies in our results. Also I did not understand from the article with what time horizon the prediction of traffic on the network was made. Let me explain with a concrete example. We used traffic data that was obtained also from [PeMS] District 4. The data was collected by 64 sensors on the Los Angeles Highway for 4.5 months in 2016 with a frequency of 5 minutes. We also used Mean Absolute Error(MAE), Root Mean Square Error(RMSE) and Mean Absolute Percentage Error(MAPE) to evaluate the prediction performance and got next results for one hour prediction:

Vector autoregression: MAE(mph)-7.26, MAPE(%)-14.15, RMSE(mph)-10.83;
FC-LSTM: MAE(mph)-3.52, MAE(mph)-7.86, RMSE(mph)-6.87;
GCGRU: MAE(mph)-4.92, MAE(mph)-10.01, RMSE(mph)-7.58;
Our Method: MAE(mph)-3.15, MAE(mph)-7.07, RMSE(mph)-5.81;

Thus, our results are few times more accurate than those obtained by the authors of this article. I would like to understand where this difference comes from and whether there is an opportunity to eliminate it?

Author Response

Thank you for pointing out the lack of a specific description of the prediction task in our paper. Our experimental traffic flow data are recorded at 5-minute intervals. And the prediction task is the same as yours, which is also to predict the next 5-minute traffic flow by a total of 5 traffic records in the 60 minutes. We have added this part of the description to the paper based on your suggestions.

As for the large differences in our experimental results, we think that there may be differences in data quality due to different ways of data pre-processing. We remove unreasonable data, smooth burr points, and utilize the sliding window method of fixed length to fix missing data. We did not use a more reasonable and comprehensive approach for data interpolation and repair. In addition, the difference in the amount of data may also lead to the difference in the effect of the experiment. The training set of our experiment is 47 days, which is indeed less than your 4.5 months, so the training effect may be lacking.

Reviewer 2 Report

The paper proposes a codec neural network that utilizes a temporal attention mechanism to capture the deep temporal dependency. The Internal Graph Convolution Network architecture is well described.

The results prove that the Codec Network Based on Attention Mechanism is adequate and can be successfully use.

Remark: It is not clear how many instances are used in the prediction result set.

Author Response

We appreciate your pointing out the lack of a detailed description of the dataset in the paper. We use PEMSD4 and PEMSD8 datasets. PEMSD4 is collected from 307 sensors distributed on 29 roads of the San Francisco Bay area, and contains 56 days from 1/1/2018 to 2/28/2018; PeMSD8 is collected from 170 sensors distributed on 8 roads of San Bernardino, and contains 62 days from 7/1/2016 to 8/31/2018. Traffic flow data is recorded every five minutes. We split each dataset and selected 47 days as the training set and 15 days as the test set.

Round 2

Reviewer 1 Report

Ok, I think the paper can be accepted in present form